# Customers' Satisfaction of E-Banking in Bangladesh: Do Service Quality and Customers' Experiences Matter?

**Md. Abdul Bashir [1], Md. Alaul Haque [2], Aidin Salamzadeh [3,\*] and Md. Mizanur Rahman [4]**

1   Department of Computer Science & Engineering, Uttara University, Dhaka 1230, Bangladesh;
    bashirupm@yahoo.com
2   Department of Business Administration, Metropolitan University Sylhet, Sylhet 3104, Bangladesh;
    alaul@metrouni.edu.bd
3   Faculty of Management, University of Tehran, Tehran 1411713114, Iran
4   BRAC Business School, BRAC University, Dhaka 1212, Bangladesh; mizanmgt@gmail.com
\*   Correspondence: salamzadeh@ut.ac.ir

**Abstract:** The banking sectors are optimistic that electronic banking (E-banking) will help them provide better customer service and strengthen customer relationships. Despite this, a relatively low priority has been given to the level of satisfaction that E-banking users in Bangladesh have regarding the quality of the services they receive and their overall experiences. Consequently, this study aims to determine the effect of service quality and customer experiences on the level of satisfaction perceived by E-banking customers in Bangladesh. Using a convenience sampling technique and a self-administered questionnaire, we gathered data from 315 E-banking customers. The independent variable (service quality and customer experience) and dependent variable (customer satisfaction) on a five-point "Likert-Type Scale" explain the degree to which participants agree or disagree with the questionnaire's statements. Covariance-based structural equation modeling (CB-SEM) was utilised to analyse the gathered data. The findings of this study indicate that service quality and customer experience significantly positively affect E-banking customer satisfaction in Bangladesh. The outcomes of this study will urge the banking authorities to prioritize service quality to boost customer satisfaction by suggesting several steps to improve the efficiency, effectiveness, and security of the E-banking system.

**Keywords:** Bangladesh; commercial banks; customer' satisfaction; customer' experience; E-banking; service quality

## 1. Introduction

Unlike every other service industry, the banking sector, with the aid of science and technological innovation, interacts closely with their customers, who have become more critical than ever before. It is acknowledged that the banking industry is the one that is primarily affected by the Internet and developments in information communication technology (ICT). However, electronic banking (E-banking) services started their journey in Bangladesh in 2001 [1]. This development has meant that Bangladesh currently has 61 scheduled banks operating throughout the country [2]. And, the banking sector is very competitive [3–6]. In Bangladesh, the total number of internet users is 112.72 million, and the total number of E-banking customers is 3.38 million. Bangladesh's Internet banking users account for 2.99% [7]. However, very few users/clients take the services from the E-banking processes in Bangladesh due to different factors such as fearing personal account security, complications of online sites banks, poor website response, etc. [1].

E-banking allows consumers to safely and easily access their bank accounts. Information technology has advantages in cost-efficiency, flexible maintenance, promptness, efficacy, and dramatically increased productivity [8–11]. Customers can securely and instantly access and monitor their finances online, regardless of location or time. Furthermore,

Internet banking allows users to verify their current balance from a remote location through the Internet. E-banking enables customers to manage their accounts from anywhere and at any time, regardless of whether it is a holiday, evening, or night. In E-banking, the customer's opinion is critical in determining the service's quality. Certain consumers value the quality and responsiveness of service providers above all else, while others value privacy and protection above all else. Some customers also value the website's design and ease of use, demonstrating that different customers have different needs and expectations from E-banking service providers. Nevertheless, some common circumstances can be found in the customer's intentions, which are legitimate expectations that ensure the study's interest. Furthermore, previous studies showed that, compared to internal factors, customers were susceptible to behavioral factors when it came to customer service delivery through E-banking [1]. As a result, there is a clear link between behavioral and E-banking technologies regarding consumer engagement and satisfaction with E-banking. As a result, the extent of E-banking adoption would directly impact consumers' satisfaction with service delivery regarding behavioral factors [1].

In the context of E-banking services, previous research has revealed (dis)advantages related to client attitudes and documented reasons why customers detest utilizing E-banking services. The literature on what inspires customers to adopt digital banking services [1,12] and how to keep them happy is primarily focused on E-banking services [13]. Many factors or constraints could be to blame for the limited adoption of electronic banking services [14]. Not all customers attempt E-banking services, for example, because of network issues with local phone services [15], a lack of trust in service safety [14,16], a lack of reading abilities and consciousness [6], and the mentality associated with traditional banking services [17]. Financial inclusion, trustworthiness, technology use and culture, gender inequality, and religion were all explored in a previous study on the acceptability of mobile banking [18]. Furthermore, the need for complete client safety, security, and quick access to electronic banking is becoming a growing source of worry [9,19–23]. The primary cause was the banks' lack of service quality, including the proper IT infrastructure and security mechanisms [24–28]. Service quality is an explanation of disparities between expectations and capabilities in addition to quality factors. Reliability, responsiveness, tangibility, communication, trustworthiness, security, and competence help customers evaluate service quality [29]. Contrarily, the degree to which a consumer's product or service matches their expectations is referred to as customer satisfaction [30]. However, most of Bangladesh's E-banking issues stem from a lack of infrastructure and customers' satisfaction (CS). However, to find the research gaps, the recent works concerning E-banking in Bangladesh are shown in Table 1.

**Table 1.** Research gaps of E-banking in Bangladesh.

| Authors | Independent Variable | Dependent Variable |
|---|---|---|
| [31] | Service quality, web design and content, security and privacy, convenience, and speed | Customer satisfaction |
| [32] | Review paper | |
| [24] | E-banking services | Financial performance |
| [33] | Review paper | |
| [34] | Conceptual paper | Conceptual paper |
| [35] | Security and loyalty | E-Banking practices |

Source: Authors' creation.

After a critical evaluation of Table 1, it was found that the majority of the research conducted on the conventional banking system in Bangladesh is definitely based on the reviewing literature, and some authors have focused on the direct effects of E-banking services on customers' satisfaction (CS). Still, less attention has been given to customer experience (CE). Consequently, this study aims to determine the effect of service quality (SQ) and customer experience (CE) on the level of satisfaction perceived by E-banking customers in Bangladesh.

The rest of the paper is put together as follows. The following section discusses the study's theoretical framework and hypotheses. After that, the overall methodology is described. The study also explains how the data were processed and what was discovered. Finally, the implications of the study are discussed.

## 2. Research Framework and Development of Hypotheses

In 1985, Davis [36] developed the Technology Adoption Model (TAM) to determine how successfully people will accept new technologies, ideas, or products. TAM describes why individuals accept new technology or new ideas, as well as how external factors or variables (SQ and CE) influence people's feelings about new products or systems when they adopt them [37]. The model also reveals how individuals perceive new products or systems after adoption [37]). According to studies (such as Venkatesh and Davis, 1996 [38]), individuals are more likely to acquire a new piece of technology or system if they perceive that it will benefit them and be easy to use. A few studies' findings support the abovementioned arguments [39]. There is evidence to suggest that many psychological and sociological factors have been added to the original TAM in order to gain a better understanding of how people's actions and thoughts may change as a result of adopting new technology or an invention. However, the following research framework (Figure 1) has been developed based on the assumptions of the TAM model.

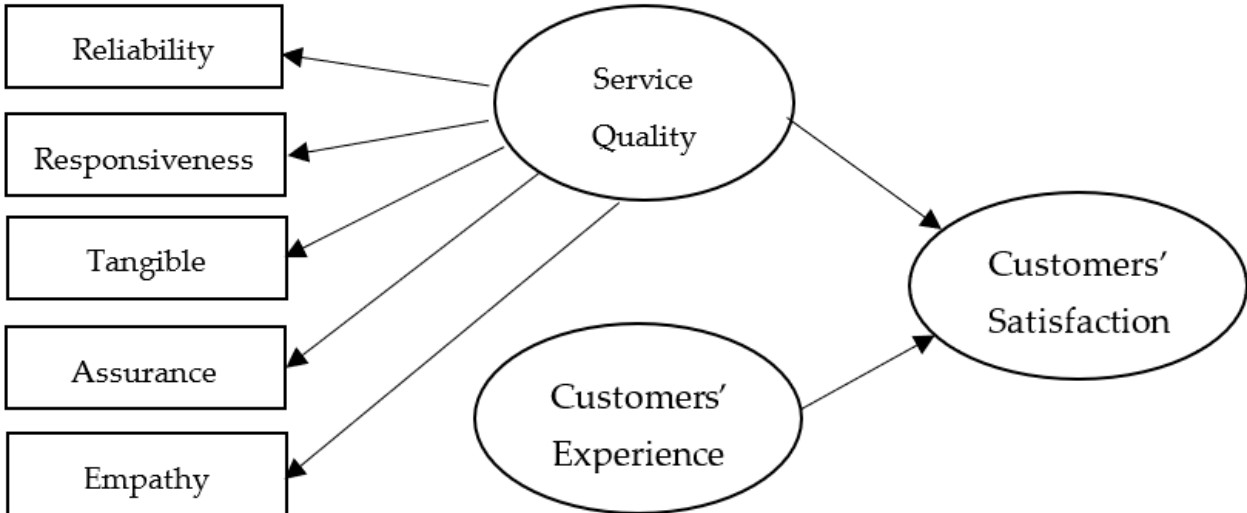

**Figure 1.** Theoretical framework.

Many researchers have given their extensive attention and focus on their research to identify the close connections between CS and SQ, and their studies empirically evaluated the validation that SQ is a very necessary or key tool that is widely used to measure CS [29,40–44]. In the same way, Zeithaml et al. (2011) [45] identified that dissatisfaction or satisfaction is a service evaluation offered to fulfil a consumer's expectations or needs. Moreover, service quality is a necessary antecedent of CS, and auspicious quality of perceived service consigns higher consumer satisfaction and vice versa [46] even in current modern banking and financial sectors [47–49]. Depending on the literature review, focusing on service quality with the customer or client satisfaction in the banking sector context, quality of service is identified to be an obligatory proceeding of CS.

**H1.** *Service quality positively affects customers' satisfaction.*

Mamat, Haron, and Razak (2014 [50]) suggested that to develop a positive customer experience, the service provider must be very intimate with the customers, and the interrelationship between them should be very efficient so that customers can recieve the same services they require from the company. If all those criteria are met, customers will have a certain experience with the service, which will drive customer loyalty and satisfaction.

Most of the studies show that customer experiences and customer satisfaction are positively correlated [50–53]. Numerous studies indicate that customers tend to be more satisfied with an organisation, particularly when they have a more positive experience with the company [51,54,55]. Based on the aforementioned statements, the researchers formulate the subsequent hypothesis.

**H2.** *Customers' experiences positively affect customer satisfaction.*

### 3. Research Methodology

#### 3.1. Sampling Technique and Sample Size

The selection of sampling technique and sample size is one of the crucial tasks for researchers [56]. The respondents of this research are E-banking users in Bangladesh. With the convenience sampling technique, a questionnaire was distributed to about 400 E-banking users to collect data. We distributed the questionnaire to the respondents of three divisions (Dhaka, Sylhet, and Rajshahi) of Bangladesh. We started the data collection on 3 November 2022 and closed the data collection after three months and one day (4 February 2023). Customer satisfaction is the dependent variable, while service quality and customer experience are independent variables. "The assumption of theory testing" justifies convenience sampling in this research. According to Zafar et al. (2022) [57], convenience sampling is a good choice for research that tests theory assumptions by constructing a study framework. There were 335 completed questionnaires returned. There were 20 completed questionnaires deleted due to respondents' missing data (9) and outliers (11). Finally, 315 E-banking users met the sampling criterion for AMOS statistical studies [58].

#### 3.2. Research Variables with Sources

This study uses questionnaires as a research tool. The questionnaire consists of two parts: (a) the demographic part and (b) the main research variables part with sources (Table 2). All items concerning each variable were scored on a five-point Likert scale [59].

**Table 2.** Measurement scale with sources.

| Variable | Dimensions | Measures | Sources | Reliability |
|---|---|---|---|---|
| Service Quality | 5 | 25 | [25,29] | 0.89 |
| Customers' Experience | 1 | 7 | [25,29] | 0.91 |
| Customers' Satisfaction | 1 | 10 | [60,61] | 0.90 |

#### 3.3. Statistical Tools

To do the preliminary analysis (missing values, normality, and multicollinearity), the researchers have used SPSS-21 version software, while to test the formulated hypotheses, AMOS software 21 has been used. Covariance based-SEM (CB-SEM) using AMOS is useful to test the theoretical assumptions [62,63].

### 4. Analyses and Findings

We have divided our analyses into three categories, i.e., a. a preliminary analysis, b. a descriptive and demographic analysis, and c. a measurement and structural model analysis.

#### 4.1. Preliminary Analysis

We checked the missing values, outliers, normality tests, and multicollinearity in the preliminary analysis part. There were 20 completed questionnaires deleted due to respondents' irregular or missing data (9) and outliers (11).

*4.2. Demographic and Descriptive Findings*

4.2.1. Demographic Findings

Regarding respondents that took part in this research (see Table 3), 59% are male while 41% are female; additionally, 35% of the participants are married while 65% are not. On the other hand, 75% of respondests were in the 26 to 33 age range, 55% were private employees, and 70% completed their postgraduate degree. Lastly, 69% of respondents had an income level between BDT 30,000 and 50,000.

**Table 3.** Demographic information.

| Demographic Information | Findings |
|---|---|
| Gender | Male = 59%<br>Female = 41% |
| Age | 18–25 = 15%<br>26–33 = 75%<br>Above 34 = 10% |
| Level of education | High school and college = 5%<br>Undergraduate = 14%<br>Postgraduate = 70%<br>Other = 11% |
| Occupation | Government employees = 14%<br>Private employees = 55%<br>Businessmen = 15%<br>Housewife = 6%<br>Students = 10% |
| Maritalstatus | Married = 35%<br>Unmarried = 65% |
| Income (per month) | Less than BDT 20,000 = 5%<br>Between BDT 20,000 to 30,000 = 8%<br>Between BDT 30,000 to 50,000 = 69%<br>Between BDT 50,000 to 80,000 = 11%<br>Above BDT 80,000 = 7% |

4.2.2. Descriptive Findings

Table 4 shows that the mean value of CS is 3.7093, and SD is 0.90202. It indicates that the CS regarding E-banking in Bangladesh is neutral. The correlation findings indicate that SQ (r = 0.556 **) and CE (0.690 **) have significant (**) positive relationships with CS.

**Table 4.** Descriptive statistics.

| | Mean | SD | A_CS | A_Sev_Qul | A_CE | Collinearity Statistics Tolerance | VIF |
|---|---|---|---|---|---|---|---|
| A_CS | 3.7093 | 0.90202 | 1 | | | | |
| A_Sev_Qul | 3.7664 | 0.73055 | 0.556 ** | 1 | | 0.718 | 1.392 |
| A_CE | 3.1556 | 1.05766 | 0.690 ** | 0.531 ** | 1 | 0.718 | 1.392 |

*4.3. Measurement and Structural Models*

4.3.1. Measurement Model

According to Mahajan (2021) [64], all research constructs have achieved unidimensionality because each item's factor loading was more significant than 0.6. The correlation values between the latent exogenous concepts are less than 0.90, which shows that discriminant validity is not an issue. But when the fitness indexes meet the criteria GFI > 0.9, CFI > 0.9, RMSEA $\leq$ 0.085, and the ratio of chi sq/df was less than 5.00, the concept validity was attained (Figure 2 and Table 5). We used the structural model to analyze the direct influence in the Analysis and Results section.

**Table 5.** Results of model fit.

|  | Absolute Fit | | Incremental Fit | | Parsimonious Fit |
|---|---|---|---|---|---|
|  | RMSEA | GFI | AGFI | CFI | Chisq/df |
| Measurement Model | 0.056 | 0.844 | 0.819 | 0.940 | 2.00 |

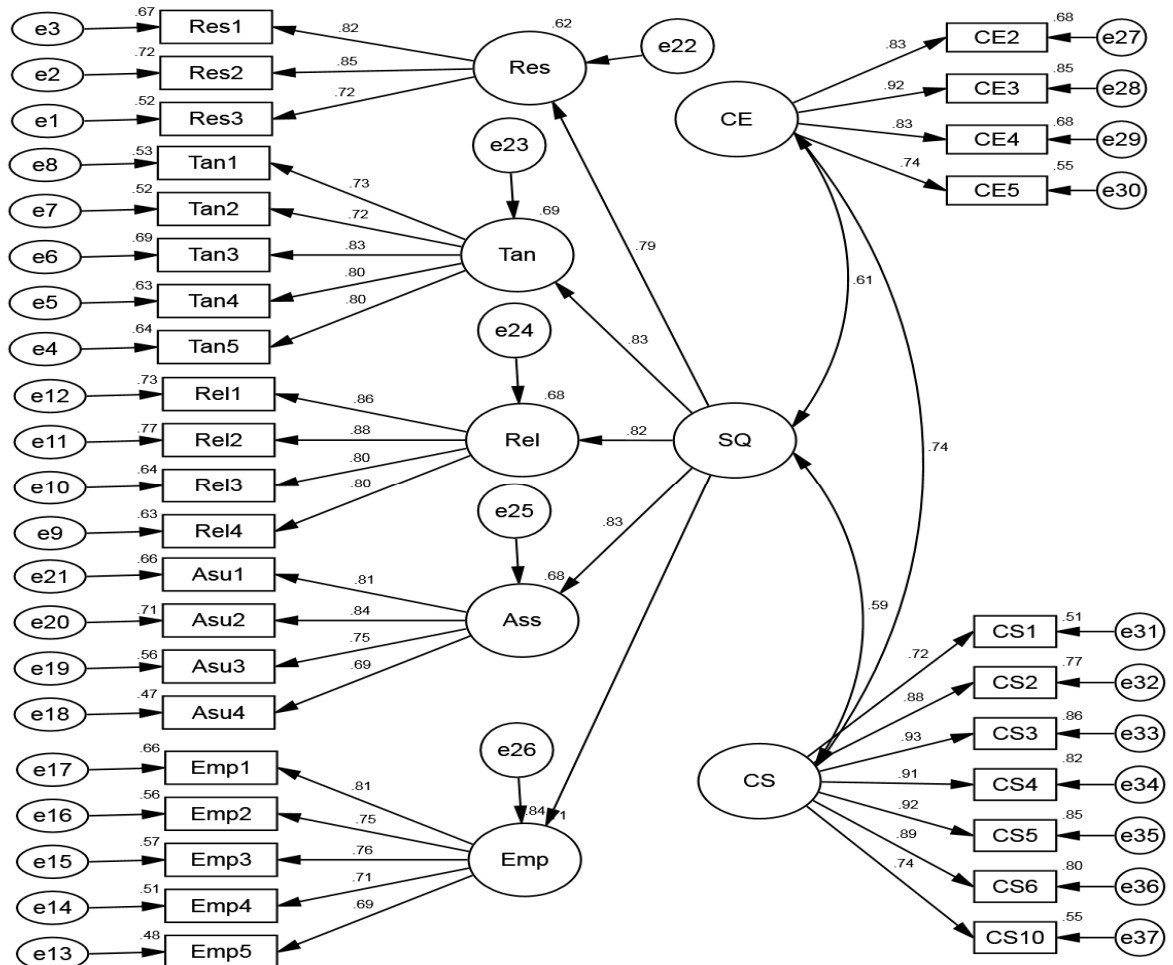

**Figure 2.** Measurement model.

### 4.3.2. Structural Model

Table 6 and Figure 3 are shown below:

**Table 6.** Results of the direct standardized effect.

| Variables |  | Variables | Estimate | S.E. | C.R. | *p* |
|---|---|---|---|---|---|---|
| SQ | --- | CS | 0.23 | 0.057 | 4.834 | ***(significant) |
| CE | --- | CS | 0.60 | 0.043 | 10.261 | *** (significant) |

To understand the direct impact of SQ on CS, refer to Figure 3 and Table 6. Overall, service quality and customer experience explained 58% of the variation in CS (R2 = 0.58). Results showed that SQ and CE positively significant impact CS (βSQ: = 0.23 and *p* = 0.000 ** and βCE: = 0.60 and *p* = 0.000 **, respectively). H1 and H2 are therefore supported.

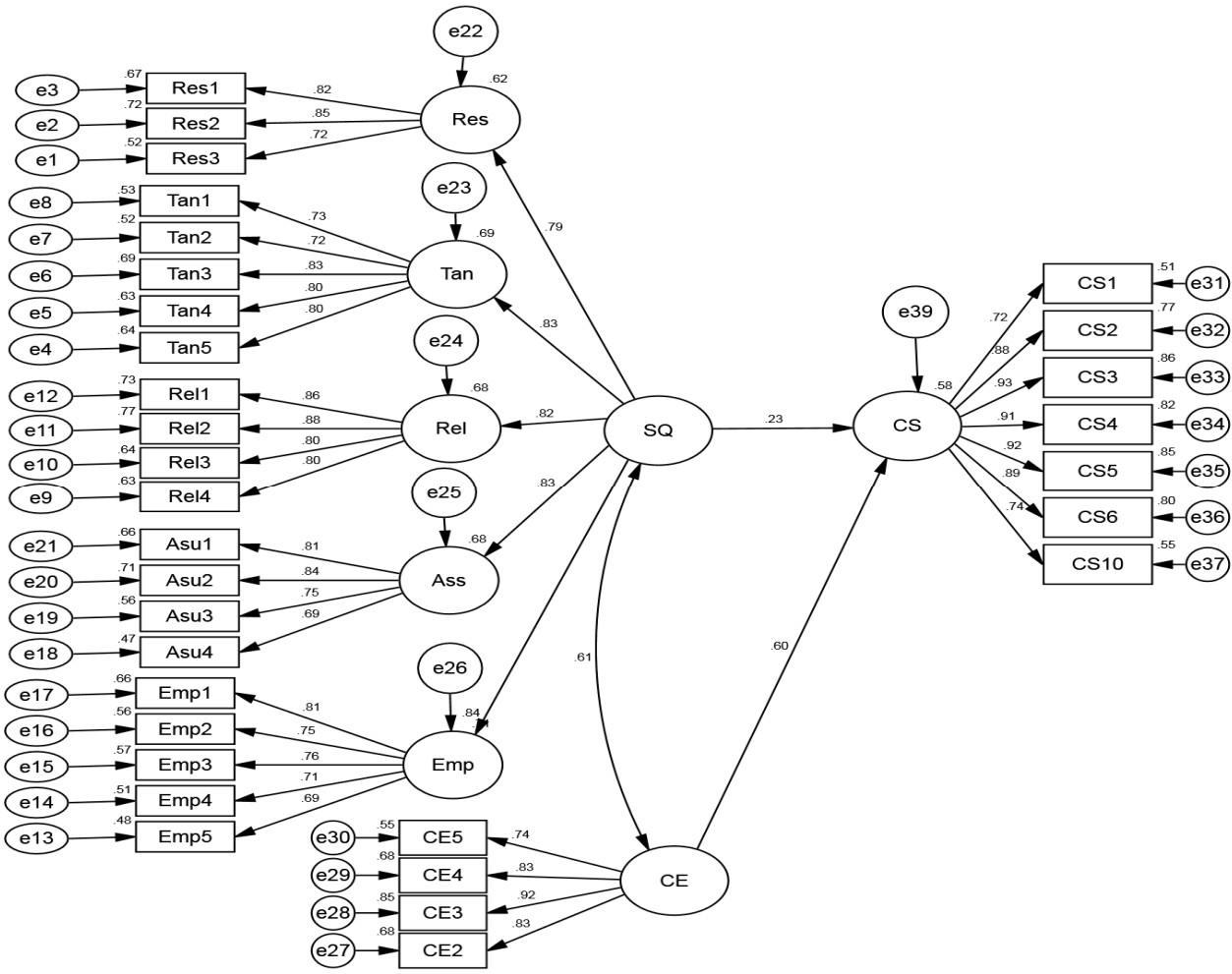

**Figure 3.** Structural model.

## 5. Discussion of the Findings

This section explains the study's discussion based on the findings. Our findings showed that both SQ and CE positively impacted CS. Our first finding is in line with those obtained by prior researchers [65–67]. In addition, Khan and Fasih (2014) [68] carried out research on the banking industry in Bangladesh and discovered that SQ has a substantial association with CS. The findings of that study are in line with the findings of the current study. Banks remain strong in their commitment to service their consumers despite the intense competition in the industry. The service is offered at every hour (24/7) and at a single location, all to provide superior service quality and keep existing customers. Customers in Bangladesh demand the service level they receive, and when unsatisfied with it, they go elsewhere (other banks). The service quality model suggests superior service is the best way to keep loyal consumers. The assumptions of TAM also support this finding.

Our second finding showed that CE significantly positively affects the CS of E-banking in Bangladesh. It means customers are satisfied if they have a positive or better experience. Various earlier studies found a similar result that CE has a positive relationship with CS [69–73]. Customer experience in banking services, especially in electronic banking services, is the interaction between banks (employees) and account holders over a long period. The customer experience that Bangladeshi customers earn is the bank's attraction, service discovery, and service rendering flow [28,74,75]. Account holders or E-banking users in Bangladesh are not very technologically sound. The process of adopting online banking sometimes becomes difficult for them. Therefore, the perceived ease of use of E-banking is considered very crucial. Customers experiencing easy-to-use

E-banking activities such as opening transactions, transferring money to other accounts, and paying various bills are delighted. Hassle-free, easy, and comfortable operation banking activities ensure a practical and attractive customer experience. E-banking is becoming very popular in Bangladesh. Banks ensure a good customer experience so that customers become satisfied and loyal to them. To provide better customer experience and service, banks offer various facilities to solve their problems.

## 6. Contributions

### 6.1. Theoretical Contributions

This study investigated the relationship between SQ and CE with CS. The statistical analyses indicated that SQ and CE have significant effects on CS. These findings were an empirical confirmation of the Technology Acceptance Model (TAM). Regarding theoretical implications, this study is one of the leading studies in Bangladesh to show the significance of a significant association between SQ, CE, and CS. Second, covariance-based Structural Equation Modeling (CB-SEM) is utilized to statistically validate the model for generating critical dimensions of CS implementation. Most of the researchers had not considered CE as a determinant of the satisfaction level of E-banking users in their past studies. However, this study evaluated the CE as a determinant variable to measure the CS of E-banking users, which is supported by the assumptions of the TAM model.

### 6.2. Practical Contributions

The SQ, CE, and CS implementation discussed in this study would be benefit both to present and future E-banking customers in Bangladesh. They could utilize service quality and customer satisfaction implementation findings in their bank industry. It saves time and cost for E-banking users to implement the right tools with the proper methods. In addition, not only the IV of DV implementation is shown in the research, but the significant relationship of those variables is well defined; therefore, it is easier and simpler to achieve CS in the Bangladeshi bank industry by utilizing the E-banking tools and the CS model in a proper sequence as suggested in this study. This research clarifies the interrelationships among these variables. The second practical implication shown in this study is that when the customer satisfaction model is implemented effectively, and indeed in the Bangladeshi bank industry, it will provide outstanding results for the Bangladeshi bank industry.

## 7. Conclusions and Areas for Further Research

The banking industry is hopeful that E-banking will help them give better customer service and build stronger ties with their customers [10,11,76,77]. Even so, the level of satisfaction that E-banking users in Bangladesh have with the quality of the services they receive and their general experiences has yet to be given much attention. Therefore, the purpose of this study is to determine how the degree of satisfaction of E-banking clients in Bangladesh is affected by both the quality of the service provided and the consumer's overall experience. We have adopted a quantitative technique to fulfill the research objectives. The results of this study show that customer satisfaction with E-banking in Bangladesh is significantly favourably impacted by service quality and customer experience. The results of this study will convince the authorities in the banking industry to place a more excellent value on the fulfillment of their customers. The outcomes of this study will urge the banking authorities to prioritise service quality to boost customer satisfaction by suggesting several steps to improve the efficiency, effectiveness, and security of the E-banking system. This study is based on convenience sampling, including non-probability sampling from the selected divisions in Bangladesh. Further studies could be carried out more scientifically with a probability sample and a statistically significant sample size. The researchers also propose surveying a different part of the country to improve the findings' generalizability.

**Author Contributions:** Conceptualize, M.A.B.; data collection, M.A.B.; analysis, M.M.R.; methodology, M.A.H.; discussions, A.S.; limitations and future direction, M.A.H.; referencing, M.A.H.; overall guidelines and proof reading, A.S. All authors have read and agreed to the published version of the manuscript.

**Funding:** This research received no external funding.

**Institutional Review Board Statement:** During the data collection, the primary purpose was clearly written on the consent sheet.

**Informed Consent Statement:** During the data collection, the primary purpose was clearly written on the consent sheet.

**Data Availability Statement:** Available on request.

**Acknowledgments:** We are grateful to the respondents who willingly provided the data.

**Conflicts of Interest:** The authors declare no conflict of interest.

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
