# Peer review of "Customers’ Satisfaction of E-Banking in Bangladesh: Do Service Quality and Customers’ Experiences Matter?"

_fintech, doi:10.3390/fintech2030036_

Round 1

Reviewer 1 Report

Thank you for preparing the article. I have the following minor comments:

1- Please proof edit the article.

2- You have to consider articles from different countries to develop the research problem in the introduction part. In this regard, you can benefit from the following articles to follow their style. (Note: it is not necessary to cite them/ all of them)

1. Al-Sartawi, A., Reyad, S., and Madbouly, A. (2022). Shariah, presentation and content dimensions of Web 2.0 applications and the firm value of Islamic financial institutions in the GCC countries. Journal of Islamic Marketing. Vol.13, No 9, 1988-2005.

2. Al-Sartawi, A., Sanad, Z., Momany, M.T., Al-Okaily, M. (2023). Accounting Information System and Islamic Banks’ Performance: An Empirical Study in the Kingdom of Bahrain. In: Musleh Al-Sartawi, A.M.A., Razzaque, A., Kamal, M.M. (eds) From the Internet of Things to the Internet of Ideas: The Role of Artificial Intelligence. EAMMIS 2022. Lecture Notes in Networks and Systems, vol 557. Springer, Cham.

3. Al-Okaily, M., Al-Sartawi, A., Hannoon, A., & Khalid, A. A. (2022) Information Technology Governance and Online Banking in Bahrain. In: Musleh Al-Sartawi A.M.A. (eds), Artificial Intelligence for Sustainable Finance and Sustainable Technology. ICGER 2021. Lecture Notes in Networks and Systems, vol 423. Springer, Cham.

4. Al-Sartawi, A., Al-Okaily, M., Hannoon, A., & Khalid, A. A. (2022) Financial Technology: Literature Review Paper. In: Musleh Al-Sartawi A.M.A. (eds), Artificial Intelligence for Sustainable Finance and Sustainable Technology. ICGER 2021. Lecture Notes in Networks and Systems, vol 423. Springer, Cham.

5. Gupta, N. (2019). Influence of demographic variables on synchronisation between customer satisfaction and retail banking channels for customers of public sector banks of India. International Journal of Electronic Banking, 1(3), 206-219.

6. Shihadeh, F. (2020). Online payment services and individuals' behaviour: new evidence from the MENAP. International Journal of Electronic Banking, 2(4), 275-282.

7. Prakash, A., Mahajan, Y., & Gadekar, A. (2022). Adoption of mobile money among internal migrant workers during the corona pandemic in India: a study focused on moderation by mode of payments. International Journal of Electronic Banking, 3(2), 144-162.

8. Bhambra, R. K. (2022). Adoption of online banking in Goa amidst the pandemic. International Journal of Electronic Banking, 3(2), 163-175.

9.   Kumar, H., & Sofat, R. (2022). Digital payment and consumer buying behavior-an empirical study on Uttarakhand, India. International Journal of Electronic Banking, 3(4), 337-357.

3-You can use the previous studies to enhance the practical and theoretical implications of your study.

4- The results need to be compared with the previous studies.

Please proof edit the article.

Author Response

Reviewers comment’

Authors’ response

Please proof edit the article.

Thank you for your comment, we have edited the article.

You have to consider articles from different countries to develop the research problem in the introduction part. In this regard, you can benefit from the following articles to follow their style. (Note: it is not necessary to cite them/ all of them)

1. Al-Okaily, M., Al-Sartawi, A., Hannoon, A., & Khalid, A. A. (2022) Information Technology Governance and Online Banking in Bahrain. In: Musleh Al-Sartawi A.M.A. (eds), Artificial Intelligence for Sustainable Finance and Sustainable Technology. ICGER 2021. Lecture Notes in Networks and Systems, vol 423. Springer, Cham.

2. Al-Sartawi, A., Al-Okaily, M., Hannoon, A., & Khalid, A. A. (2022) Financial Technology: Literature Review Paper. In: Musleh Al-Sartawi A.M.A. (eds), Artificial Intelligence for Sustainable Finance and Sustainable Technology. ICGER 2021. Lecture Notes in Networks and Systems, vol 423. Springer, Cham.

3. Gupta, N. (2019). Influence of demographic variables on synchronisation between customer satisfaction and retail banking channels for customers of public sector banks of India. International Journal of Electronic Banking, 1(3), 206-219.

4. Shihadeh, F. (2020). Online payment services and individuals' behaviour: new evidence from the MENAP. International Journal of Electronic Banking, 2(4), 275-282.

5. Prakash, A., Mahajan, Y., & Gadekar, A. (2022). Adoption of mobile money among internal migrant workers during the corona pandemic in India: a study focused on moderation by mode of payments. International Journal of Electronic Banking, 3(2), 144-162.

6. Bhambra, R. K. (2022). Adoption of online banking in Goa amidst the pandemic. International Journal of Electronic Banking, 3(2), 163-175.

7.   Kumar, H., & Sofat, R. (2022). Digital payment and consumer buying behavior-an empirical study on Uttarakhand, India. International Journal of Electronic Banking, 3(4), 337-357.

Thank you for providing the insightful articles; We have cited 7 articles.

You can use the previous studies to enhance the practical and theoretical implications of your study

We have done it, I mean, we have classified our implications into two (theoretical and practical implications).

The results need to be compared with the previous studies

We have done it in the discussion part.

Reviewer 2 Report

I have read the article "Customers' Satisfaction of E-banking Users in Bangladesh: Do Service Quality and Customer Experience Matter?" (fintech-2543690), which is an interesting reading beacuse of its framework simplicity and linearity in carrying out the empirical demonstration.

In this specific regard, I have still two concerns:

1. there is no contextualisation at all. Thus, I would have appreciated a better description of the payment-methods landscape in Bangladesh. For instance: how large/recurrent is the use of e-banking and digital payments in general? How did these data evolve over time? And what are the main payment instruments used? I think a table/figure containing all these information might be of great utility to better contextualise the authors' analysis;

2. the article itself is very short for being a research article. I would strongly suggest to critically re-read all sentences and ask whether some additional elements might be added (or not).

Some sentences are difficult to understand an/or sound "non-English" (for instance, "[t]he researcher also proposes...") and/or are somehow "exaggerated" (for instance, "[t]he results of this study will convince the authorities..."). In my opinion, while re-reading the entire text, such critical parts should be amended/adapted/changed.

Author Response

Reviewers comment’

Authors’ response

I have read the article "Customers' Satisfaction of E-banking Users in Bangladesh: Do Service Quality and Customer Experience Matter?" (fintech-2543690), which is an interesting reading because of its framework simplicity and linearity in carrying out the empirical demonstration. In this specific regard, I have still two concerns:

Thank you so much for your nice comment.

1. There is no contextualisation at all. Thus, I would have appreciated a better description of the payment-methods landscape in Bangladesh. For instance: how large/recurrent is the use of e-banking and digital payments in general? How did these data evolve over time? And what are the main payment instruments used? I think a table/figure containing all these information might be of great utility to better contextualise the authors' analysis;

We have done it based on your comment. The contextualization part has been added in the introduction part.

2. The article itself is very short for being a research article. I would strongly suggest to critically re-read all sentences and ask whether some additional elements might be added (or not).

We have added a few parts and elements. Now, the total number of words is over 6000.

Reviewer 3 Report

I find the paper topic interesting relating to customer satisfaction of E-banking users.

The literature covered highlight that authors know deeply the research topics.

The methodologies are appropriate to the proposed objective of research paper and I suggested to added commentary  of the items of questionnaires  in order to understand better the results of the model.

The conclusion are well presented ans suggest to include the limits of the research.

Author Response

Reviewers comment’

Authors’ response

I find the paper topic interesting relating to customer satisfaction of E-banking users.

Thank you so much for your nice comment.

The literature covered highlight that authors know deeply the research topics.

Thank you so much for your nice comment.

The methodologies are appropriate to the proposed objective of research paper and I suggested to added commentary of the items of questionnaires in order to understand better the results of the model.

This is a very good comment. Based on your comment, we have added our questionnaire as Appendix One. Now, the readers will get a clear idea regarding our items.

The conclusion are well presented and suggest to include the limits of the research.

Thank you so much for your nice comment.

Reviewer 4 Report

An interesting research topic, especially from a practical point of view. It is worth presenting the research hypotheses in the introduction, apart from presenting the aim of the research. Research tools, especially statistical ones, need to be described in more detail. You need to specify the conditions of their use and indicate their strengths and weaknesses. Conclusions should be extended with an assessment of the research methods used. The economic conclusions resulting from the conducted research should be extended, what is the significance of this and for whom (banks, bank customers, etc.) ?

Author Response

Reviewers comment’

Authors’ response

An interesting research topic, especially from a practical point of view. It is worth presenting the research hypotheses in the introduction, apart from presenting the aim of the research. Research tools, especially statistical ones, need to be described in more detail. You need to specify the conditions of their use and indicate their strengths and weaknesses. Conclusions should be extended with an assessment of the research methods used. The economic conclusions resulting from the conducted research should be extended, what is the significance of this and for whom (banks, bank customers, etc.)?

Thank you so much for your nice comment.

We have added the research methods and implications in the conclusion part.

Reviewer 5 Report

Dear Authors,

I commend your efforts on this paper. I know it is difficult to write a scientific article. Your paper provides some important insights, however in its present form, it needs a significant revision before it can be published in a scientific journal. There are a number of important issues. First, your study lacks even the most basic descriptive analysis of a statistical survey. There is no coherent reason/explanation for choosing your methodological approach. It lacks basic information such as the date of your survey or the method of administering the survey. There is no information on the questions or scale used. The discussion is not really a scientific discussion of findings. The figures are missing several words. The referencing is inconsistent throughout, mixing up a name-based system with a number based one. There are significant problems with the structure and coherence. It lacks overall scientific soundness and the findings are doubtful at best.

Significant proofreading and editing required.

Author Response

Reviewers comment’

Authors’ response

I commend your efforts on this paper. I know it is difficult to write a scientific article. Your paper provides some important insights, however in its present form, it needs a significant revision before it can be published in a scientific journal. There are a number of important issues. First, your study lacks even the most basic descriptive analysis of a statistical survey. There is no coherent reason/explanation for choosing your methodological approach. It lacks basic information such as the date of your survey or the method of administering the survey. There is no information on the questions or scale used. The discussion is not really a scientific discussion of findings. The figures are missing several words. The referencing is inconsistent throughout, mixing up a name-based system with a number based one. There are significant problems with the structure and coherence. It lacks overall scientific soundness and the findings are doubtful at best.

Thank you so much for your nice comment.

We have added the descriptive analysis part.

We have added our used items/scale in the appendix (one) part.

We have modified our discussion part.

We have changed our referencing as followed the journal’s instructions.

Round 2

Reviewer 5 Report

The current version is a significant improvement on the previous one. However, some minor issues remain. There is still no information on the precise dates of the survey, and the paper needs a table presenting the demographic characteristics of the survey respondents.

Also, I mentioned the need for a measurement scale in my previous comment. The authors have provided the survey questions. But by measurement scale I meant the authors need to provide the source of the questions, not just the questions themselves. Where did you get the questions from? Where they adapted from a scale or previous study? Kindly add this information to the paper please.

Minor proofreading

Author Response

Reviewer’s comments’

Authors’ response

The current version is a significant improvement on the previous one. However, some minor issues remain. There is still no information on the precise dates of the survey, and the paper needs a table presenting the demographic characteristics of the survey respondents.

Thank you for your nice comment.

However, we have given the information (precise date) of our data collection.

In addition, we have also added the demographic information as Table. 

Also, I mentioned the need for a measurement scale in my previous comment. The authors have provided the survey questions. But by measurement scale I meant the authors need to provide the source of the questions, not just the questions themselves. Where did you get the questions from? Where they adapted from a scale or previous study? Kindly add this information to the paper please.

Thank you for your great concern, we have added a measurement scale Table (Table 2) with necessary sources.